# Research on Transportation Mode Recognition Based on Multi-Head Attention Temporal Convolutional Network

**DOI:** 10.3390/s23073585

**Published:** 2023-03-29

**Authors:** Shuyu Cheng, Yingan Liu

**Affiliations:** College of Information Science and Technology, Nanjing Forestry University, Nanjing 210037, China

**Keywords:** transportation mode recognition, deep learning, temporal convolutional network, multi-head attention mechanism

## Abstract

Transportation mode recognition is of great importance in analyzing people’s travel patterns and planning urban roads. To make more accurate judgments on the transportation mode of the user, we propose a deep learning fusion model based on multi-head attentional temporal convolution (TCMH). First, the time-domain features of a more extensive range of sensor data are mined through a temporal convolutional network. Second, multi-head attention mechanisms are introduced to learn the significance of different features and timesteps, which can improve the identification accuracy. Finally, the deep-learned features are fed into a fully connected layer to output the classification results of the transportation mode. The experimental results demonstrate that the TCMH model achieves an accuracy of 90.25% and 89.55% on the SHL and HTC datasets, respectively, which is 4.45% and 4.70% higher than the optimal value in the baseline algorithm. The model has a better recognition effect on transportation modes.

## 1. Introduction

With the rapid development of mobile internet technology and the advancement of technology, smartphones are becoming increasingly indispensable in people’s daily lives. Many sensors equipped on smartphones are mainly used to process and record information. This data information can be used to effectively monitor people’s daily behavior and identify people’s transportation modes [1,2,3,4].

Transportation mode recognition is a judgment of the current transportation mode of the user, which is a considerable branch of people’s activity recognition. Daily transportation modes include: stationary, walking, running, bicycle, bus, car, train and subway. Users often use different means of transportation during travel and have different travel needs. These requirements require intelligent mobile terminals to determine in advance the transportation modes within the user’s location. Transportation mode recognition is a fundamental problem that plays a crucial role in several fields. Transportation mode detection can help individuals avoid congested routes and have a comfortable transportation experience. It is also beneficial for transportation planning and management departments to carry out urban road planning and vehicle scheduling and solve the problem of transportation congestion. Furthermore, it can also quickly arrange the most suitable driving plan for ambulances.

To date, researchers have proposed machine learning algorithms to solve transportation mode recognition problems, such as decision tree (DT) [5], random forest (RF) [6,7,8], support vector machine (SVM) [9], etc. Nick et al. [10] used a plain Bayesian classifier and a support vector machine to preprocess the sensor data and extract features manually for transportation mode recognition. Hemminki et al. [11] preprocessed the collected dataset and gravity estimation, manually performed feature extraction and finally placed the extracted features in a classifier to identify the transportation mode. Ashqar et al. [6] use a two-layer framework that employs machine learning techniques, including a k-nearest neighbor, classification and regression tree, support vector machine and random forest. The framework combines the newly extracted features with traditional time-domain features to form a feature pool, improving classification accuracy. These traditional machine learning algorithms have certain drawbacks, i.e., they require specialized domain knowledge to extract features manually, which can affect the accuracy of classification on the one hand, and on the other hand, they can cause a large workload due to the difficult feature design.

Deep learning algorithms can effectively solve the above problems, i.e., they can autonomously learn the intrinsic laws and potential features of data, improve efficiency and enhance recognition accuracy. As a result, researchers started to shift from traditional machine learning algorithms to deep learning algorithms [12,13,14], such as convolutional neural network (CNN) [15,16,17], recurrent neural network (RNN) [18,19], long short-term memory network (LSTM) [20,21], etc. Liu et al. [22] proposed an end-to-end bi-directional LSTM-based classification framework to classify users’ trajectories into different modes of transportation. Qin et al. [23] first used convolutional neural networks to learn features and then used LSTM to further extract features from the CNN output. Features are further extracted using LSTM, which ultimately leads to an improved accuracy of transportation mode recognition. Sharma et al. [24] used deep learning networks, recurrent neural networks and convolutional neural networks to learn time-related mode information, which performed well on the validation dataset. Gong et al. [25] proposed a convolutional neural network-based approach to identify subways, trains and buses with high accuracy and showed good robustness.

However, these deep learning algorithms still have some shortcomings: recurrent neural network computation does not support parallelism and has high training overhead. The convolutional neural network can only extract short-time local features due to perceptual wilderness. In addition, the existing methods do not assign reasonable weights to the extracted potential features, and the algorithms only show good recognition effects on a single small-scale dataset with insufficient generalization ability.

This paper proposes a novel transportation mode recognition algorithm consisting a multi-head attention (MHA) mechanism, temporal convolutional network (TCN) and convolutional neural network (CNN), with the following main contributions:We leverage the temporal convolutional networks to the transportation feature learning on individual sensor data. The temporal convolutional network uses inflated convolution to increase the perceptual field of view and learn the long-time dependent features of the sensor data. Simultaneously, it is trained with parallel computation and short-time overhead.We adopt the multi-headed attention mechanisms to extract multiple spatial features. Compared with single-headed attention, the multi-headed attention model assigns more moderate weights to the features and highlights the vital feature information. It has high identification accuracy for similar modes of transportation, such as trains and subways.Our proposed algorithm was validated on SHL [26] and HTC [27] datasets and compared with machine learning algorithms (DT, RF, SVM) and deep learning algorithms (LSTM, CNN, CNN + LSTM, MSRLSTM). The experimental results show that the TCMH model significantly improves the accuracy, precision, recall and F1-score classification metrics compared with the above algorithms.

The rest of this paper is organized as follows: Section 2 introduces the overall architecture of the TCMH model and explains the basic principle of the algorithm. Section 3 describes two datasets and evaluation metrics and shows the experimental results of the TCMH model. Finally, Section 4 summarizes the work of this paper.

## 2. Algorithm

This paper proposes a TCMH model for transportation mode recognition. The model mainly consists an input layer, a temporal convolutional network layer, a multi-headed attention layer, a convolutional neural network layer and an output layer, and the overall architecture is shown in Figure 1 as follows:Input layer: Multiple sensor data are input to the input layer after normalization and its output is used as the input of the temporal convolutional neural network.Temporal convolutional layer (TCN): A network structure is superior to recurrent neural networks and convolutional neural networks, consisting causal convolution, expansion convolution and residual connectivity.Multi-head attention layer (MHA): The output of the TCN is used as the input of this module. The features acquired by each head are fused so that the final developed features can represent global dependencies.Convolutional layer (CNN): this network consists a convolutional layer with a convolutional kernel size of 64, a maximum pooling layer and a global average pooling layer.Output layer: it consists a fully connected layer with neurons of eight and a Softmax activation function. The maximum subscript of neurons is used as the final output of transportation mode classification, i.e., eight transportation mode classification results.

### 2.1. Input Layer

The sensor data collected by smartphones changes with time and is a typical time series data. Ten sets of data, linear acceleration sensors X, Y and Z axes, gyroscope sensors X, Y and Z axes, geomagnetic sensors X, Y and Z axes and barometric sensors are selected and processed through the input layer to obtain 10 tensors of $\left(B,500,1\right)$ size. Where B is the number of samples selected for each training, which is set to 32. 500, and is generated using a non-overlapping sliding window segment with a sampling frequency of 100 Hz at 5 s. One refers to the specific features used for transportation mode recognition, such as linear acceleration X-axis data features.

### 2.2. Temporal Convolutional Layer

A recurrent neural network (RNN) is the preferred neural network in processing time series data, which can reflect the relationship between the current moment and the previous moment information and has certain short-term memory capabilities. As variants of recurrent neural networks (long short-term memory networks (LSTM) and gated recurrent neural networks (GRU)), they can solve the problems of gradient explosion and small memory capacity of recurrent neural networks. However, it also has the disadvantage of processing data serially and having high computational overhead.

Temporal convolutional neural networks can effectively solve the above problems. Firstly, the TCN can easily obtain stable gradients, which can avoid the gradient explosion problem to a certain extent. Secondly, it can extract time-dependent features and increase memory capacity by increasing the perceptual field size. Furthermore, it can perform large-scale parallel computation, accelerate the computation speed and improve the computation efficiency.

Temporal convolutional neural networks include the following concepts: causal convolution, expansion convolution and residual connection. Causal convolution only utilizes the sensor time series data before that moment and does not focus on the data information after that moment. Thus, it can solve the information leakage problem of relying on future data at that moment. Since causal convolution can only focus on the sensor time data of the preceding shorter moments, if we want to obtain information features on long-time scales, we need to add expansion convolution. Expansion convolution obtains more feature information by injecting voids into the convolution. The dilation convolution has a hyperparameter dilation, which refers to the number of intervals performed during sampling. The hyperparameter dilation=1 indicates that the sample is required for each data point. Dilation=2 suggests that the sample is performed every two data points, and so on. The causal expansion convolutional structure is shown in Figure 2. Adding residual connections in TCN can avoid the loss of transportation mode features due to the deepening of network layers, thus ensuring that the transportation mode recognition accuracy does not drop significantly.

The number of filters used in this paper is 32. Therefore, a data tensor with the input of $\left(B,500,1\right)\;$ can obtain a feature tensor of size $\;\left(B,250,32\right)$ after the temporal convolutional neural network and maximum pooling. At the same time, the internal features of the eight transportation modes with long-time dependencies are fully explored to improve the training efficiency when the sensor input data are used.

### 2.3. Multi-Head Attention Layer

In recent years, the attention mechanism has been widely used [28,29,30] and has become one of the research hotspots in deep learning. It uses weight size to measure different feature information when processing data information, providing a larger weight to important features and a smaller weight to relatively unimportant features. It improves the efficiency of feature learning and can dig out more valuable implicit information from the massive data. However, the ordinary attention mechanism only extracts the sensor data feature dependencies from one dimension, which can only learn the feature information with limitations. In view of this, the multi-headed attention mechanism is introduced to solve this problem.

The multi-headed attention mechanism first maps the input into b different subspaces through a fully connected layer (FC). Each subspace contains a query matrix $Q_{j}$, key matrix $K_{j}$ and value matrix $V_{j}$, where j=1,…,h. Then, the attention calculation is performed in parallel in the b subspaces using the scaled dot-product attention function, and the attention calculation formula is shown in Equation (1).
(1)$$h_{j}=Attention\left(Q_{j}\;,K_{j}\;,V_{j}\right)=Softmax\left(\frac{Q_{j}K_{j}^{T}}{\sqrt{d}}\right)V_{j}$$
where $h_{j}$ denotes the attention value of the *j*th space and d represents the dimension of the key.

Finally, the obtained attention values are stitched together and the output can be obtained after the matrix $W_{o}$.
(2)$$W_{O}\left[\begin{array}{c} h_{1}\\ \vdots \\ h_{b} \end{array}\right]$$
where $W_{o}$ is the matrix of learnable parameters. 

The schematic diagram of the multi-headed attention structure is shown in Figure 3. According to the above principle, the output result x of TCN is passed through the multi-head attention module to make the final extracted data feature information more comprehensive, which is helpful in improving the accuracy of transportation mode recognition. The input and output of this layer are all feature tensors of size $\left(B,250,32\right)$. 

### 2.4. Convolutional Layer

The convolutional neural network is a feed−forward neural network proposed by researchers inspired by the concept of perceptual wilderness. The convolutional neural network is good at mining local features in a small range and extracting characteristic values of targets and has strong applicability. It is used for target recognition and classification in complex and diverse environments [31,32]. It has three properties: local connection, weight sharing and pooling. Local connection means that the neurons in the nth layer are connected to only some neurons in the (n − 1)th layer, and only local features are extracted. Weight sharing means that the neurons in the previous layer are scanned with a convolution kernel (the values in the convolution kernel are called weights), i.e., the same set of weights is used to convolve the neurons in the previous layer. The 1D convolution example in Figure 4 exemplifies the two properties of local connection and weight sharing. The role of pooling is to perform feature selection and reduce the number of transportation mode features. Maximum pooling is chosen, which reduces the number of neurons used in the transportation mode recognition network while maintaining the constancy of the local features of the fused data.

In this paper, 10 sensors are first stitched together, and then a convolutional network with a convolutional kernel of 64 is used for local feature extraction. The maximum pooling is used to select beneficial features for improving the accuracy of transport mode recognition. Where poolsize=2, which in turn yields a feature tensor of size $\;\left(B,125,64\right)$. The global average pooling is calculated by averaging the 64 transportation mode data feature maps obtained through the convolutional neural network, which can reduce the dimensionality of the output and prevent overfitting. After averaging pooling, the feature tensor of the maximum pooling output $\left(B,125,64\right)$ becomes a tensor of $\left(B,64\right)$.

### 2.5. Output Layer

Since there are eight transportation mode labels in the dataset, the number of neurons in the last fully connected layer is set to eight; then, the Softmax activation function (the function can compress the data range of each neuron in the range of 0 to 1, and the sum of all data is 1) is used to output the probabilities corresponding to the eight transportation modes. Finally, the position corresponding to the maximum probability is used as the final result of transportation mode classification.

## 3. Experiments and Analysis

### 3.1. Datasets

Here, experiments are conducted on two public datasets to evaluate the performance of the TCMH model:SHL dataset. The dataset was recorded in 2017 by three volunteers who placed a Huawei Mate 9 phone on a part of their body, and it took the volunteers more than 7 months. The SHL dataset contains 272 h of sensor data. The SHL dataset can be used to analyze transportation conditions and estimate satellite coverage, which this paper uses for transportation mode recognition. Eight transportation modes are classified as still, walk, run, bike, car, bus, train and subway. Ten sets of data are selected from the dataset as raw data: three-axis linear acceleration, three-axis gyroscope, three-axis magnetometer and barometric pressure sensor.HTC dataset. The HTC dataset was collected by more than 100 volunteers using HTC phones in 2012. It contains nine sensor types: acceleration sensors on the X, Y and Z axes, geomagnetic sensors on the X, Y and Z axes and gyroscopic sensors on the X, Y and Z axes. The dataset contains 8311 h of sensor data. Unlike the SHL dataset, the HTC dataset classifies transportation modes into 10 categories. Two transportation categories, motorcycle and high-speed rail, were dropped to maintain consistency between the two datasets.

### 3.2. Data Preprocessing

Since different sensor timing data have different dimensions, the final recognition effect will be affected if not processed. To eliminate the influence of the magnitude, improve the convergence speed of the model and to increase the recognition accuracy, we use the Z-score normalization method to operate on the data. The data processed by this method conform to the standard normal distribution. The formula of Z-score normalization is as follows:(3)$$X^{\prime }=\frac{X-u}{\sigma }$$
where u is the mean of the original data used for mode recognition and σ is the standard deviation of the original data used for mode recognition in the dataset.

To better evaluate the model effect, the two datasets, SHL and HTC, are divided into training, validation and test sets, respectively, and the allocation ratio is 3:1:1.

### 3.3. Metrics

To verify the effectiveness of the TCMH model, we use the accuracy rate as the main index to evaluate the model. We use precision, recall and F1-score to assess the recognition effect of eight transportation modes.

Accuracy is used to describe the proportion of correctly predicted samples to all samples, i.e., the proportion of correctly classified transportation mode samples to all samples used for transportation mode classification, as shown in Equation (4):(4)$$Accuracy=\frac{\sum_{j=1}^{k}{TP}_{j}}{N}$$
where *k* is the number of classified transportation modes, *N* is the total number of all experimental samples and ${TP}_{j}$ is the number of samples correctly classified by transportation mode *j*.

Precision is relative to the classification prediction results of transportation mode and describes the proportion of samples with correct positive predictions to all samples with positive predictions, as shown in Equation (5):(5)$$Precision=\frac{{TP}_{j}}{{TP}_{j}+{FP}_{j}}$$
where ${FP}_{j}$ is the number of samples that misclassify other transportation modes as mode *j*.

Recall is the proportion of samples correctly predicted as positive to all actual positive samples relative to the transportation mode classification samples, as shown in Equation (6):(6)$$Recall=\frac{{TP}_{j}}{{TP}_{j}+{FN}_{j}}$$
where ${FN}_{j}$ is the number of samples that misclassify transportation mode *j* as other modes.

The F1-score is determined by both precision and recall, as shown in Equation (7):(7)$$F1-Score=\frac{2\times recall\times precision}{recall+precision}$$

### 3.4. Experimental Configuration

We adopt the Keras deep learning framework to train the TCMH model. The Adam optimizer (learning rate is set to 0.001) is selected. For the multi-classification problem, the cross-entropy loss function is selected. The number of training epochs is set to 100, and the batch size is set to 32. The experimental configuration is shown in Table 1.

### 3.5. Experimental Comparison of Different Algorithms

The RF, DT, SVM, CNN, LSTM, CNN + LSTM and MSRLSTM [13] are used as the baseline algorithms to evaluate the performance of our proposed TCMH model. Among these baselines, CNN is a part of the proposed model in this paper, and CNN + LSTM is composed of the above algorithms, CNN and LSTM. Three machine learning algorithms, RF, DT and SVM, are implemented using Sklearn. The detailed parameters of the baseline algorithms are shown in Table 2.

The accuracy of each algorithm on the SHL and HTC datasets is shown in Figure 5 and Figure 6. According to the experimental results, the following conclusions can be drawn: Deep learning algorithms show a higher recognition effectiveness than machine learning algorithms. This is due to the ability of deep learning algorithms to learn deep potential features from the sensor temporal data, which are more helpful for transportation mode classification. Among the three machine learning algorithms, compared to DT and SVM algorithms, RF reduces the risk of overfitting by building many trees and has the highest recognition accuracy on SHL and HTC datasets, which are 77.23% and 82.27%, respectively. Among the deep learning algorithms, the TCMH model outperforms other baseline algorithms. This is because the temporal convolutional network included in the TCMH model can capture more transportation mode information without losing information features, and the multi-headed attention mechanism can fuse the features so that the final acquired features have a global view. The accuracy of the TCMH model exceeds the other algorithms on both the SHL dataset and HTC dataset at 90.25% and 89.55%, respectively, while the accuracy of the other algorithms in transportation mode recognition is below 86%.

For each transportation mode recognition case, the precision, recall and F1-score of each baseline algorithm and the TCMH model are shown in Table 3 and Table 4.

Table 3 and Table 4 show that the recognition effect of three transportation modes, bus, train and subway, is poor. Transportation modes’ recognition, such as running and cycling, relies only on short-time data information, while recognition of the same three transportation modes relies on longer-time data information. Each baseline algorithm is limited by the small memory capacity and short-time local features. Therefore, the precision, recall, and F1-score are all low in these three modes, with an average of about 60%. The TCMH model has good recognition results on all three transportation modes, with precision, recall and F1-score higher than 70%, reflecting the advantage of the TCMH model in recognizing transportation modes that depend on long-time information. Meanwhile, the experimental results show that the precision, recall and F1-score are higher for all algorithms when classifying the three transportation modes of walking, running and cycling. The intrinsic reason is that when people perform these three sports, there are large swaying and regular movements of the human body, which have more obvious characteristics. Although each baseline algorithm reflects good classification results on these three transportation modes, the proposed TCMH model in this paper has an advantage over the recognition results of each other baseline algorithm. All three metrics are above 89% on the SHL dataset and above 84% on the HTC dataset. In particular, the transportation mode of running achieves a precision of 100% on the SHL dataset.

### 3.6. Effect of the Number of Heads of Multi-Headed Attention Modules

The accuracy of the TCMH model is affected by the number of multi-head attention heads. To explore the optimal number of heads, we set the different numbers to observe the effect of identification on the SHL and HTC datasets, as shown in Figure 7 and Figure 8, respectively.

Figure 7 and Figure 8 show that when head = 5, the accuracy is the largest, 90.25% and 89.55% on the SHL and HTC datasets, respectively. When head = 1, there is only one head in the model, and the accuracies are 88.50% and 88.05%, respectively, which shows the advantage of multi-headed attention over single-headed attention in transportation mode recognition. When head = 12, the model has an overfitting phenomenon, and the accuracy decreases to 87.20% and 88.00%, respectively.

### 3.7. Self-Contrasting Experiments

To verify the necessity of the temporal convolutional neural network and the multi-headed attention mechanism of the TCMH model, the removal of the temporal neural network and the multi-headed attention mechanism are experimentally compared with the TCMH model, respectively. The precision, recall and F1-score are used as the metrics to measure the model.

Table 5 and Table 6 show that removing the temporal convolutional network part of the TCMH model leads to a decrease in the precision, recall and F1-score, which reflects the contribution of the temporal convolutional network to the TCMH model. For subway, on the SHL dataset, the difference between the TCMH model and TCMH model when removing the temporal convolutional network part is obvious, with 13.97%, 28.18% and 22.03% difference in precision, recall and F1-score, respectively. On the HTC dataset, the difference in precision, recall, and F1-score is 19.91%, 17.67% and 18.76%, respectively.

Table 6 and Table 7 show that the TCMH model has lower recognition results than the removal of the multi-headed attention mechanism part. However, three evaluation indexes have been improved from the overall classification results of the eight transportation modes. Especially for still, on the SHL dataset, the precision, recall and F1-score improved by 4.99%, 1.56% and 3.37%, respectively. On the HTC dataset, the precision, recall and F1-score enhanced by 3.30%, 1.99% and 2.68%, respectively.

### 3.8. Experiment of Hyperparameter Adjustment

The primary hyperparameters in the TCMH model are adjusted: the number of filters and convolutional kernel size in TCN, the dimensional value of keys in multi-headed attention, and the number of filters and convolutional kernel size in CNN.

According to the variable control method, one hyperparameter value is adjusted each time, and the results obtained are shown in Table 8 and Table 9. It can be seen that the adjustment of hyperparameters has a certain influence on the recognition precision of transportation modes. In particular, the precision of recognizing trains on the SHL and HTC datasets differed by 11.39% and 8.91%, respectively. 

## 4. Conclusions

This paper proposes a novel transportation mode recognition model, TCMH. By combining TCN and MHA, the accuracy of transportation mode recognition is increased, and the training process is speeded up. The TCMH algorithm is also energy efficient, using only the multiple lightweight sensors integrated in the smartphone to detect transport patterns. The experimental results on two datasets show that the proposed model is significantly better than baseline algorithms such as the RF-, DT-, SVM-, CNN-, LSTM-, CNN + LSTM- and MSRLSTM-based transportation mode algorithms. It also confirms the reasonable scalability of TCMH.

There are some limitations in the TCMH model. The accuracy of recognition can be further improved, and the complexity of the model can be further reduced. In future scientific work, we will continue to research deep learning models with lower computational overhead and higher recognition accuracy, and further improve transportation mode recognition performance in practical application scenarios.

## Figures and Tables

**Figure 1 sensors-23-03585-f001:**
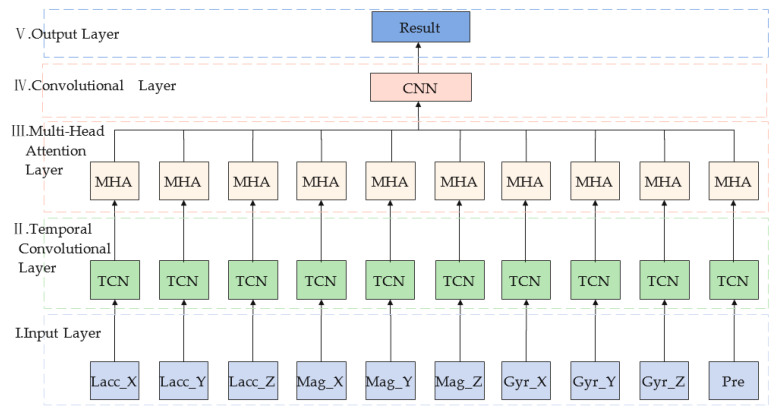
Schematic diagram of the TCMH model framework.

**Figure 2 sensors-23-03585-f002:**
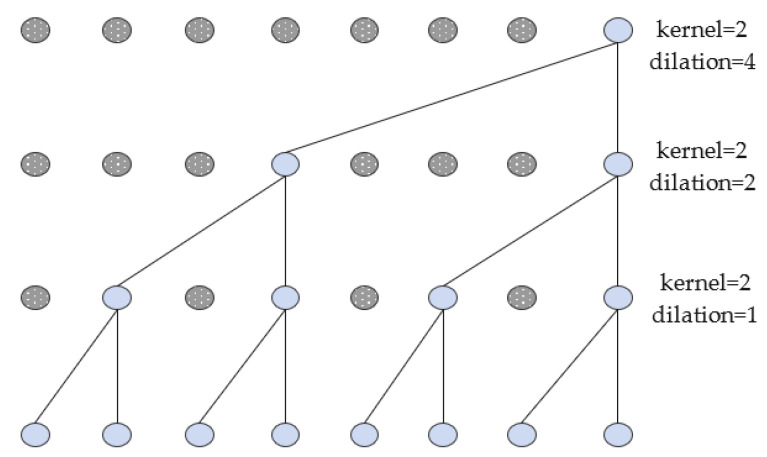
Causal expansion convolution structure diagram.

**Figure 3 sensors-23-03585-f003:**
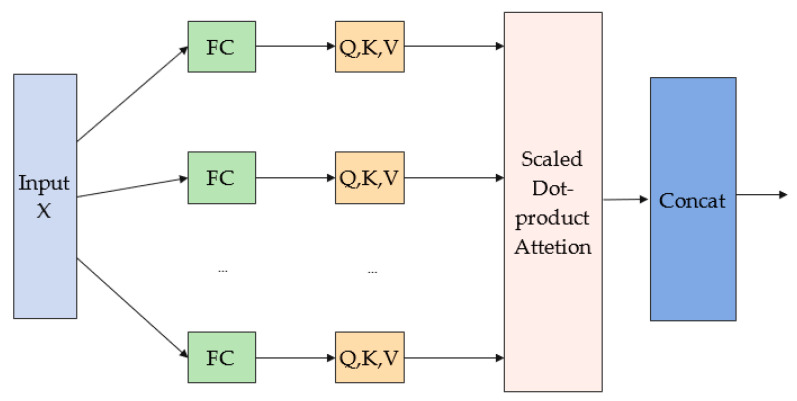
Schematic diagram of multi-headed attention structure.

**Figure 4 sensors-23-03585-f004:**
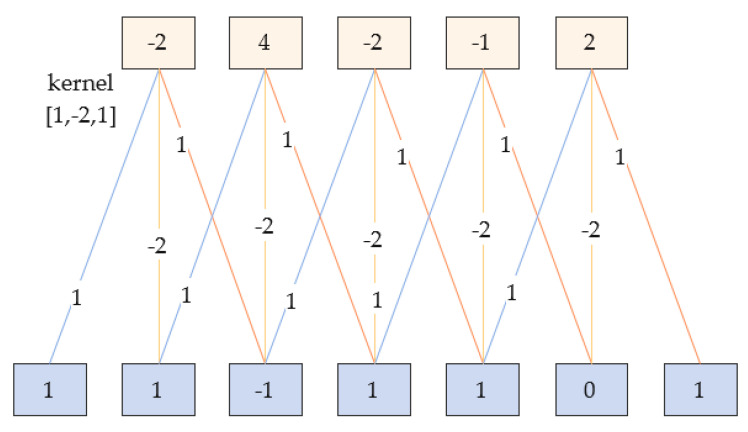
Schematic diagram of the one—dimensional convolution.

**Figure 5 sensors-23-03585-f005:**
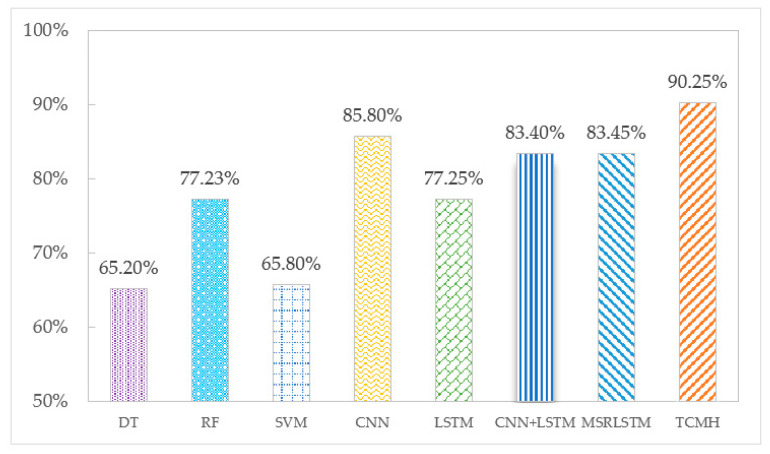
Accuracy of each algorithm on the SHL dataset.

**Figure 6 sensors-23-03585-f006:**
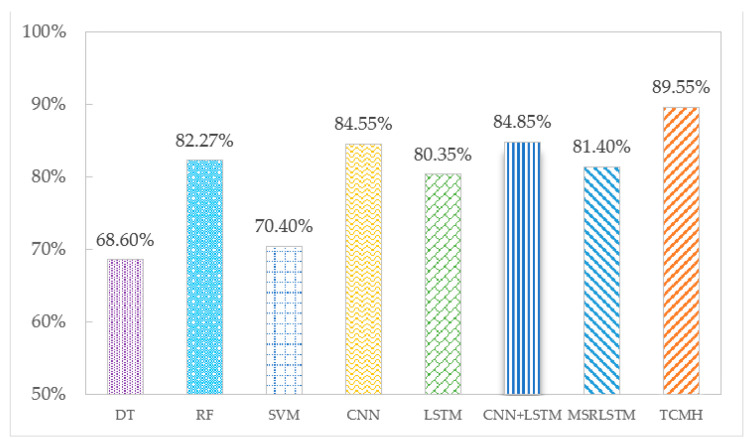
Accuracy of each algorithm on the HTC dataset.

**Figure 7 sensors-23-03585-f007:**
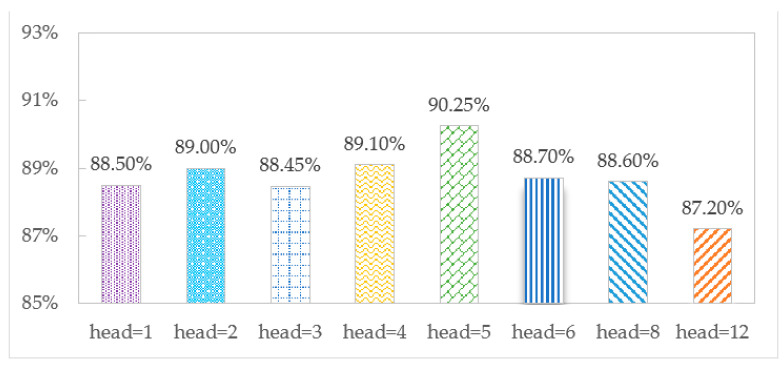
Accuracy comparison for different numbers of heads on the SHL dataset.

**Figure 8 sensors-23-03585-f008:**
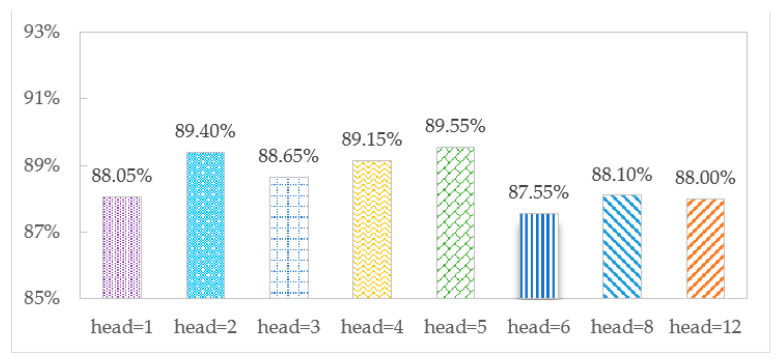
Accuracy comparison for different numbers of heads on the HTC dataset.

**Table 1 sensors-23-03585-t001:** Experimental configuration.

Name	Configuration
CPU	Intel(R) Xeon(R) CPU @ 2.20 GHz
Memory	16 G
GPU	Tesla P100
Operating System	Ubuntu 18.04.6
Python Environment	Python 3.7.15
TensorFlow Version	2.9.2

**Table 2 sensors-23-03585-t002:** Detailed parameters of the baseline algorithms.

Name	Architecture
DT	criterion = gini
RF	n_estimators = 50
SVM	Kernel = rbf
CNN	C(64)- C(128)-GAP- FC(8)-Softmax
LSTM	LSTM(128)-FC(8)-Softmax
CNN + LSTM TCMH	C(64)-LSTM(128)-FC(8)-Softmax TCN(32)-MHA(5)-CNN(64)-FC(8)-Softmax

Note: FC denotes fully connected layer; C denotes convolutional neural network; GAP denotes global average pooling.

**Table 3 sensors-23-03585-t003:** Comparison of evaluation metrics of different algorithms on the SHL dataset.

Algorithm	Metrics	Still	Walk	Run	Bike	Car	Bus	Train	Subway
	Precision	61.26%	84.32%	94.05%	78.10%	62.96%	47.34%	48.94%	45.89%
DT	Recall	60.15%	81.89%	96.13%	78.55%	61.15%	49.38%	51.45%	43.54%
	F1-score	60.70%	83.09%	95.08%	78.32%	62.04%	48.34%	50.16%	44.68%
	Precision	75.79%	91.15%	97.25%	88.82%	73.21%	63.23%	64.01%	63.14%
RF	Recall	79.69%	91.86%	97.79%	87.54%	83.21%	60.49%	66.15%	48.35%
	F1-score	77.69%	91.50%	97.52%	88.18%	77.89%	61.83%	65.06%	54.76%
	Precision	46.74%	84.10%	93.07%	83.96%	58.71%	45.77%	63.95%	62.28%
SVM	Recall	84.83%	86.09%	96.41%	77.39%	65.47%	36.73%	45.43%	31.23%
	F1-score	60.27%	85.08%	94.71%	80.54%	61.90%	40.75%	53.12%	41.60%
	Precision	85.61%	93.04%	99.59%	94.42%	88.19%	80.80%	74.29%	68.60%
CNN	Recall	90.27%	96.58%	98.77%	93.62%	87.55%	78.02%	75.09%	64.55%
	F1-score	87.88%	94.78%	99.17%	94.02%	87.87%	79.39%	74.69%	66.51%
	Precision	83.98%	87.94%	99.58%	91.89%	69.50%	64.65%	61.06%	61.46%
LSTM	Recall	75.49%	94.30%	97.12%	86.81%	80.95%	59.91%	66.79%	53.64%
	F1-score	79.51%	91.01%	98.33%	89.28%	74.79%	62.19%	63.79%	57.28%
	Precision	79.24%	92.86%	100.00%	91.60%	83.81%	81.31%	68.99%	68.78%
CNN + LSTM	Recall	89.11%	93.92%	98.35%	92.77%	85.35%	75.00%	71.48%	59.09%
	F1-score	83.88%	93.38%	99.17%	92.18%	84.57%	78.03%	70.21%	63.57%
	Precision	84.07%	95.02%	100.00%	88.98%	84.25%	82.41%	68.31%	63.51%
MSRLSTM	Recall	88.33%	94.30%	98.77%	92.77%	84.25%	76.72%	70.04%	60.91%
	F1-score	86.15%	94.66%	99.38%	90.83%	84.25%	79.46%	69.16%	62.18%
	Precision	89.47%	93.82%	100.00%	96.12%	95.47%	91.23%	81.02%	74.09%
TCMH	Recall	92.61%	98.10%	98.77%	94.89%	92.67%	89.66%	80.14%	74.09%
	F1-score	91.01%	95.91%	99.38%	95.50%	94.05%	90.43%	80.58%	74.09%

**Table 4 sensors-23-03585-t004:** Comparison of evaluation metrics of different algorithms on the HTC dataset.

Algorithm	Metrics	Still	Walk	Run	Bike	Car	Bus	Train	Subway
	Precision	75.82%	72.26%	92.31%	72.73%	73.07%	40.22%	48.70%	56.73%
DT	Recall	79.91%	77.50%	93.66%	76.45%	70.94%	42.53%	46.79%	49.87%
	F1-score	77.81%	74.79%	92.98%	74.54%	71.99%	41.34%	47.72%	53.08%
	Precision	92.36%	80.65%	95.22%	80.45%	78.88%	80.00%	82.59%	76.03%
RF	Recall	83.59%	86.50%	97.07%	85.67%	90.99%	50.57%	66.07%	74.31%
	F1-score	87.76%	83.47%	96.14%	82.98%	84.50%	61.97%	73.41%	75.16%
	Precision	72.33%	74.59%	93.36%	71.94%	63.10%	56.41%	55.28%	81.91%
SVM	Recall	74.51%	79.25%	96.10%	76.11%	84.64%	12.64%	39.29%	58.19%
	F1-score	73.40%	76.85%	94.71%	73.96%	72.30%	20.66%	45.93%	68.04%
	Precision	91.64%	89.71%	96.95%	89.27%	85.03%	71.30%	74.77%	74.91%
CNN	Recall	87.09%	85.16%	93.38%	93.85%	88.63%	62.10%	79.21%	76.32%
	F1-score	89.30%	87.37%	95.13%	91.50%	86.79%	66.38%	76.92%	75.61%
	Precision	84.36%	78.44%	92.70%	78.61%	80.71%	70.48%	79.40%	75.58%
LSTM	Recall	85.76%	82.42%	93.38%	81.03%	87.86%	59.68%	78.22%	61.65%
	F1-score	85.06%	80.38%	93.04%	79.80%	84.13%	64.63%	78.80%	67.91%
	Precision	86.71%	86.45%	94.78%	87.38%	84.20%	79.00%	83.05%	78.76%
CNN + LSTM	Recall	90.73%	84.77%	93.38%	92.31%	90.37%	63.71%	72.77%	76.69%
	F1-score	88.67%	85.60%	94.07%	89.78%	87.17%	70.54%	77.57%	77.71%
	Precision	83.23%	87.93%	92.81%	84.83%	83.21%	58.74%	75.26%	77.13%
MSRLSTM	Recall	88.74%	79.69%	94.85%	91.79%	85.93%	67.74%	72.28%	64.66%
	F1-score	85.90%	83.61%	93.82%	88.18%	84.55%	62.92%	73.74%	70.35%
	Precision	92.93%	92.31%	98.52%	91.54%	87.34%	73.77%	84.00%	86.80%
TCMH	Recall	91.39%	84.38%	97.79%	94.36%	94.41%	72.58%	83.17%	81.58%
	F1-score	92.15%	88.16%	98.15%	92.93%	90.74%	73.17%	83.58%	84.11%

**Table 5 sensors-23-03585-t005:** Experimental results of removing temporal convolutional network.

	SHL Dataset			HTC Dataset		
Mode	Precision	Recall	F1-Score	Precision	Recall	F1-Score
Still	75.78%	85.21%	80.22%	79.88%	85.43%	82.56%
Walk	96.77%	91.25%	93.93%	86.25%	80.86%	83.47%
Run	98.77%	98.77%	98.77%	97.73%	94.85%	96.27%
Bike	89.34%	92.77%	91.02%	87.89%	85.64%	86.75%
Car	81.89%	79.49%	80.67%	76.68%	85.55%	80.87%
Bus	73.25%	71.98%	72.61%	63.16%	48.39%	54.79%
Train	58.73%	66.79%	62.50%	72.47%	63.86%	67.89%
Subway	60.12%	45.91%	52.06%	69.58%	68.80%	69.19%

**Table 6 sensors-23-03585-t006:** Experimental results of the TCMH model.

	SHL Dataset			HTC Dataset		
Mode	Recall	F1-Score	Precision	Recall	F1-Score	Precision
Still	89.47%	92.61%	91.01%	90.45%	94.04%	92.21%
Walk	93.82%	98.10%	95.91%	92.00%	89.84%	90.91%
Run	100.00%	98.77%	99.38%	98.48%	95.59%	97.01%
Bike	96.12%	94.89%	95.50%	93.53%	96.41%	94.95%
Car	95.47%	92.67%	94.05%	87.07%	92.10%	89.51%
Bus	91.23%	89.66%	90.43%	79.82%	70.16%	74.68%
Train	81.02%	80.14%	80.58%	87.23%	81.19%	84.10%
Subway	74.09%	74.09%	74.09%	89.49%	86.47%	87.95%

**Table 7 sensors-23-03585-t007:** Experimental results of removing the multi-headed attention mechanism.

	SHL Dataset			HTC Dataset		
Mode	Precision	Recall	F1-Score	Precision	Recall	F1-Score
Still	84.48%	91.05%	87.64%	87.15%	92.05%	89.53%
Walk	92.34%	96.20%	94.23%	88.40%	86.33%	87.35%
Run	99.59%	98.77%	99.17%	98.50%	96.32%	97.40%
Bike	93.59%	93.19%	93.39%	88.89%	94.36%	91.54%
Car	91.64%	92.31%	91.97%	87.36%	91.91%	89.58%
Bus	95.10%	83.62%	88.99%	80.53%	73.39%	76.79%
Train	77.50%	78.34%	77.92%	86.81%	78.22%	82.29%
Subway	72.56%	70.91%	71.72%	88.40%	83.08%	85.66%

**Table 8 sensors-23-03585-t008:** Comparison of the precision of different hyperparameters on the SHL dataset.

Hyperparameters	Still	Walk	Run	Bike	Car	Bus	Train	Subway
TCN(32,3)d(64)CNN(64,3)	89.47%	93.82%	100.00%	96.12%	95.47%	91.23%	81.02%	74.09%
TCN(64,3)d(64)CNN(64,3)	91.03%	95.82%	100.00%	96.93%	91.27%	88.66%	69.63%	72.31%
TCN(32,2)d(64)CNN(64,3)	86.84%	95.56%	99.58%	96.55%	90.97%	91.67%	76.07%	76.80%
TCN(32,3)d(16)CNN(64,3)	90.04%	93.82%	100.00%	97.36%	90.28%	91.03%	76.87%	73.87%
TCN(32,3)d(64)CNN(32,3)	86.38%	92.45%	99.58%	95.07%	92.40%	85.65%	76.26%	62.85%
TCN(32,3)d(64)CNN(64,2)	85.24%	95.51%	100.00%	94.09%	92.91%	92.17%	72.38%	75.40%

**Table 9 sensors-23-03585-t009:** Comparison of the precision of different hyperparameters on the HTC dataset.

Hyperparameters	Still	Walk	Run	Bike	Car	Bus	Train	Subway
TCN(32,3)d(64)CNN(64,3)	90.45%	92.00%	98.48%	93.53%	87.07%	79.82%	87.23%	89.49%
TCN(64,3)d(64)CNN(64,3)	93.95%	89.87%	97.06%	88.18%	83.21%	66.10%	82.16%	82.86%
TCN(32,2)d(64)CNN(64,3)	90.06%	90.04%	97.73%	93.43%	87.16%	81.31%	89.01%	86.33%
TCN(32,3)d(16)CNN(64,3)	91.19%	87.75%	97.73%	91.67%	89.41%	67.91%	83.84%	89.02%
TCN(32,3)d(64)CNN(32,3)	90.28%	83.83%	94.85%	91.54%	88.67%	68.55%	80.10%	84.12%
TCN(32,3)d(64)CNN(64,2)	90.55%	91.90%	98.50%	93.14%	87.27%	79.09%	84.97%	85.29%

## Data Availability

The SHL dataset is publicly available in reference [27], and the HTC dataset is required to be seen in reference [8].

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
