# Peer review of "Research on Transportation Mode Recognition Based on Multi-Head Attention Temporal Convolutional Network"

_sensors, 2023, doi:10.3390/s23073585_

Round 1

Reviewer 1 Report

This paper proposed a transportation mode recognition model based on multi-headed attention temporal convolution. It is recommended to address the following comments:

1.       What is the amount of data in the two datasets?

2.       In Section 3.5, the author made an experimental comparison of different algorithms. Tabel 2 displayed detailed parameters of the baseline algorithms, but not of TCMH. As the Section 3.6 shown, the accuracy of TCMH varies with different number of head. So please show detailed parameters.

3.       In Section 3.6 to Section 3.8, why only the results of SHL dataset? Why choose this dataset?

4.       There are some minor writing edits that are needed throughout the manuscript. For example, for the classification of transportation modes, stationary, walking, running, bicycle, car, bus, train and subway are listed in Section 3.1. However, the classification in Table 3 is still, walk, run, bike, car, bus, railway and subway. I suggest that the author unify the expression. In section 3.8, “the obtained hyperparameter table is shown in 8”. Table 8?

Reviewer 2 Report

The manuscript proposes a deep learning fusion model based on a multi-head attentional temporal convolutional neural network (TCMH). The method is experimentally validated on two datasets and compared to the competitive approaches, showing promising results regarding transportation mode classification accuracy.

Below are the comments that should be addressed during the manuscript’s revision:

1.      At the end of the Introduction section, the authors should add a paragraph providing a brief overview of the manuscript’s structure.

2.      In the literature review, the authors should mention (1-2 sentences) the general state-of-the-art performance of CNNs in different applications today (besides transportation mode recognition) and illustrate by referring to the recent studies: 10.1109/ACCESS.2021.3139850, 10.3390/s22031215.

3.      Were the parameters of all machine learning and deep learning algorithms used in comparison optimized for this purpose?

4.      In the Conclusion section, the authors should state some of the limitations of the presented study.

Round 2

Reviewer 1 Report

I do not have further comments.

Reviewer 2 Report

The authors have addressed my comments.